# Modulation of working memory duration by synaptic and astrocytic mechanisms

**Sophia Becker**[1,2☉], **Andreas Nold**[2,3☉], **Tatjana Tchumatchenko**[2,3,4]*

**1** Laboratory of Computational Neuroscience, Brain Mind Institute, École Polytechnique Fédérale de Lausanne, Lausanne, Switzerland, **2** Theory of Neural Dynamics group, Max Planck Institute for Brain Research, Frankfurt am Main, Germany, **3** Institute of Experimental Epileptology and Cognition Research, Life and Brain Center, Universitätsklinikum Bonn, Bonn, Germany, **4** Institute for Physiological Chemistry, Medical Center of the Johannes Gutenberg-University Mainz, Mainz, Germany

☉ These authors contributed equally to this work.
* tatjana.tchumatchenko@uni-mainz.de

**Data Availability Statement:** The computer code underlying our results can be found in the following GitLab repository https://gitlab.rlp.net/braincodepublished/neuronal-network-simulations-June-2021.

## Abstract

Short-term synaptic plasticity and modulations of the presynaptic vesicle release rate are key components of many working memory models. At the same time, an increasing number of studies suggests a potential role of astrocytes in modulating higher cognitive function such as WM through their influence on synaptic transmission. Which influence astrocytic signaling could have on the stability and duration of WM representations, however, is still unclear. Here, we introduce a slow, activity-dependent astrocytic regulation of the presynaptic release probability in a synaptic attractor model of WM. We compare and analyze simulations of a simple WM protocol in firing rate and spiking networks with and without astrocytic regulation, and underpin our observations with analyses of the phase space dynamics in the rate network. We find that the duration and stability of working memory representations are altered by astrocytic signaling and by noise. We show that astrocytic signaling modulates the mean duration of WM representations. Moreover, if the astrocytic regulation is strong, a slow presynaptic timescale introduces a 'window of vulnerability', during which WM representations are easily disruptable by noise before being stabilized. We identify two mechanisms through which noise from different sources in the network can either stabilize or destabilize WM representations. Our findings suggest that (i) astrocytic regulation can act as a crucial determinant for the duration of WM representations in synaptic attractor models of WM, and (ii) that astrocytic signaling could facilitate different mechanisms for volitional top-down control of WM representations and their duration.

## Author summary

The ability to form memories and recall them is one of the fascinating features of our brain. Working memory operates like a memory scratch pad storing ongoing information for further processing. Here, we present a computational model dissecting the influence of astrocytes on the stability and duration of working memories. We find that a long astrocytic time constant can influence the mean duration of working memory representations

**Funding:** This study was supported by the Max Planck Society (T.T.), the German Research Foundation via CRC1080 (T.T.) and its student-postdoc cooperation program (A.N. and S.B.), the Joachim Herz Foundation (Add-on fellowship, A. N.), Studienstiftung des deutschen Volkes (S.B.). The funders had no role in study design, data collection and analysis, decision to publish, or preparation of the manuscript.

**Competing interests:** The authors have declared that no competing interests exist.

and generate a "window of vulnerability", during which some memories are tagged for long-term survival while some are terminated. The fraction of memories in the survival and termination groups could be regulated by adjusting the strength of astrocytic feedback or its time constant. This indicates that astrocytic signaling can be viewed as a candidate mechanism for top-down control of working memory representations and their duration.

## Introduction

Experimental data and computational models have provided evidence and understanding of a variety of WM correlates in the brain. These range from persistent delay activity [1–6], sequential activations of neurons in the WM population [7–9] and oscillatory WM activity [10–13] to activity-silent synaptic representations [5, 6, 14, 15]. Specifically, synaptic attractor models have reconciled experimental evidence for persistent and silent WM representations [5, 14, 16] and are thought to play a crucial role for information storage in between active states of bursting activity in the gamma and beta band [11]. Synaptic attractor models assume that information (e.g. about a WM item) is stored in the synaptic states of a neural ensemble—often through some form of short-term plasticity (STP) [17]. When a (WM) stimulus is presented to the network, the activity of the neurons causes either synaptic depression [18] or synaptic facilitation or augmentation [19], i.e. a temporary reduction or increase in synaptic efficacy, respectively. This interplay between presynaptic depression and facilitation [5, 6, 20] arises from the interaction of presynaptic vesicles and calcium dynamics [18, 21]. In contrast to Hebbian, associative forms of synaptic STP [22, 23] we consider non-associative models, i.e. in order to store information about a WM stimulus, synaptic structures to represent the stimulus in the network already need to exist. The phase space and bifurcation diagram for synaptic attractor models have been characterized in detail [24–27]. One important property that allows them to reconcile evidence of persistent and silent WM representations is that they can exhibit bistability: in certain parameter regimes, a 'silent' state (fixed point) of low, asynchronous network activity coexists with an 'active' state (limit cycle) of repetitive, synchronized spiking of the network [5, 26–28].

For sufficiently strong external inputs (or connection strengths within the network), the transition between the 'active' state (limit cycle) and the 'silent' state (saddle point) exhibits a homoclinic bifurcation [26, 27]. Then, within the bistable region, one can observe spontaneous, noise-driven transitions between the silent and the active network state if the noise is sufficiently strong to destabilize both the silent and the active state [26, 27]. Alternatively, if the silent and active states are stable with respect to a given level of noise, an increase (or decrease) in external inputs can move the network state from a silent to an active (or from an active to a silent) representation [5]. This scenario is used to model WM by interpreting the short increase of external input as a WM stimulus that induces persistent delay activity (active network state). The delay activity can be terminated by an external off-signal that switches the network back into the silent state, such as a population-specific inhibitory input [6] or a global reduction of excitatory input [5]. We also find parameter regimes where the silent state is stable with respect to noise while the active state is meta-stable, i.e. can be terminated through a noise-induced, spontaneous transition. In this case, we can switch the system to the active state by presenting a WM cue, but do not require an off-stimulus to terminate the WM representation—instead, it will terminate naturally and spontaneously by switching back to the silent state due to noise. So far, however, it is unclear whether and how this spontaneous termination of WM can be controlled.

Over the past decade, an increasing body of evidence has shown that astrocytes influence and interact with synaptic transmission at the presynaptic and postsynaptic side [29–31]. In these cases, we also speak of tripartite synapses [32]. Recent studies indicate that up to 80% of cortical synapses could be modulated in this way [33]. Astrocytic modulation of synaptic transmission can take place on timescales of several seconds [34, 35] to minutes [36, 37]. Since one astrocyte typically covers between $10^5$ synapses in mice and $2 \times 10^6$ in humans [38], astrocytic modulation is potentially relevant for the coordination of network activity [39] and could influence high-level neural computations [40]. Here, we investigate how the long timescale of a specific signaling pathway at the tripartite synapse affects the stability and duration of meta-stable active WM representations, i.e. active WM representations that are induced by a cue stimulus but terminate naturally through noise, and explain the network mechanisms underlying the observed phenomena. We (i) develop a minimal tripartite synapse model of slow astro-cytic modulation of the STP signaling pathway, and (ii) combine it with the synaptic WM model by Tsodyks [41] and Mongillo et al. [5] that is based on Tsodyks and Markram's STP model [18].

At the core of our tripartite synapse model is a recently described signaling pathway ('LPA-mechanism') that relies on the lipid messenger molecule lysophosphatidic acid (LPA), the astrocytic enzyme autotaxin (ATX), and the postsynaptic plasticity-related gene 1 (PRG1) protein [42–44]. The PRG1 protein is expressed at the postsynaptic density of glutamatergic synapses [45] in several brain regions, including the hippocampus and the somatosensory cortex [42–44], implying that PRG1-related signaling could affect tripartite synapses across large areas of the brain. The LPA-mechanism has been implicated in a shift of the excitation-inhibition balance towards hyperexcitability [42], altered sensorimotor processing [43] and schizophrenia [44, 46, 47], making it an interesting candidate signaling mechanism for our study of astrocytic modulation of WM.

We integrate our tripartite synapse model with the WM model by Tsodyks [41] and Mongillo et al. [5] and focus on the dynamical regime in which the silent state is stable and the active delay activity is metastable, i.e. can be disrupted by noise. Using firing rate and spiking network simulations of the WM model with and without astrocytic modulation of STP, we show how astrocytic signaling regulates WM representations and develop a mechanistic explanation of our findings. In particular, we answer the following three questions: (i) What role do different sources of noise have for the WM encoding, i.e. for the stability of the active WM representation? Here, we identify two mechanisms by which different sources of noise can either stabilize or destabilize active WM representations. (ii) How does astrocytic modulation of presynaptic STP impact the stability and duration of active WM representations? We characterize the distribution of durations of active WM representations, and show that its mean is continuously modulated via astrocytic regulation of synaptic efficacy. (iii) Finally, what is the specific role of the slow astrocytic timescale? We show that it allows networks with astrocyte-modulated STP to produce two qualitatively different distributions of WM durations, depending on the timecourse of astrocyte-induced stabilization and destabilization of active WM representations.

## Models and methods

### STP synapse model

Short-term plasticity (STP) is a key component of activity-silent WM models [5, 6, 14, 16, 48, 49]. Over the past ten years, mounting evidence has shown that STP is affected by astrocytic signaling via the lipid messenger lysophosphatidic acid (LPA) in the synaptic cleft [42, 43, 47]: Both the postsynaptic protein PRG1 and the astrocytic enzyme ATX modulate the LPA

concentration in the synaptic cleft, which affects binding to presynaptic LPA-receptors that in turn modulate presynaptic vesicle release probability in glutamatergic excitatory synapses [42, 43] (Fig 1A). For instance, PRG1 knockout in mice (PRG1-KO) leads to a decrease of the paired-pulse-ratio (PPR) compared to wild-type (WT) mice [47] (see Fig B in S1 Text). Building on the STP framework by Tsodyks et al. [18] and inspired by De Pitta et al.'s model of glio-modulation [31, 50], we construct a synapse model (Fig 1B) that captures the influence of this astrocytic signaling mechanism on synaptic STP.

Synaptic STP comprises temporal changes in synaptic efficacy that arise from the competing effects of neurotransmitter availability and calcium-binding at the presynaptic release sites [21]. Upon the arrival of a presynaptic spike, calcium binds at the active zones and leads to the release of neurotransmitters. In between spikes, calcium slowly unbinds and the neurotransmitter vesicles are refilled. When the depletion of neurotransmitters due to incoming spikes exceeds the rate of replenishment, then this leads to a decrease of neurotransmitter release upon presynaptic spike arrival (short-term depression). This effect is counteracted by calcium-binding that increases neurotransmitter release (short-term facilitation). These synaptic dynamics are captured by two equations (see [18] for the original formulation by Tsodyks and Markram) that describe the interaction of the presynaptic variables $x$ (amount of available neurotransmitter) and $y$ (presynaptically bound calcium):

$$\frac{dy}{dt} = \frac{U - y}{\tau_F} + U \sum_k (1 - y_k^-)\delta(t - t_k) \tag{1}$$

$$\frac{dx}{dt} = \frac{1 - x}{\tau_D} - \sum_k y_k^+ x_k^- \delta(t - t_k). \tag{2}$$

Here, $t_k$ are the arrival times of presynaptic spikes; $y_k^- = y(t_k^-), x_k^- = x(t_k^-)$ are evaluated immediately before the $k$-th spike arrival, and $y_k^+ = y(t_k^+)$ is evaluated immediately after the $k$-th spike arrival. We also refer to $r(t_k) = y_k^+ x_k^-$ as the instantaneous release of synaptic resources for the $k$-th presynaptic spike. The relative amount of ready-to-release neurotransmitter vesicles $x$ at the presynaptic active zones varies between 0 (complete depletion of vesicles) and 1 (full vesicle storage). Similarly, the relative amount $y$ of calcium that is bound in the presynaptic active zones ranges between 0 (no calcium bound) and 1 (maximum amount of calcium bound). Neurotransmitter is replenished and calcium unbinds with time constants $\tau_D = 200$ms and $\tau_F = 1500$ms, respectively (see Ref. [5], and Table E in S1 Text). The amount of presynaptically bound calcium $y$ increases with each arriving spike, proportionally to the presynaptic calcium binding rate $U \in [0, 1]$ and saturates at $y = 1$. The available neurotransmitter decreases with each transmitted spike $t_k$ by the instantaneous release $r(t_k) = y_k^+ x_k^-$ for that spike. Upon spike arrival, the synapse transmits a current of

$$I = J \sum_k y_k^+ x_k^- \delta(t - t_k), \tag{3}$$

where $J$ is the synaptic efficacy. STP synapses in rate networks are defined analogously (see section 'Firing rate network: model and parameters' in S1 Text). The major difference is that in the rate network, we consider the population-average over $x$ and $y$ instead of individual synaptic variables. In the rate model equations for $y$ (Eq. 6 in S1 Text) and $x$ (Eq. 7 in S1 Text), we therefore use the population-averaged firing rate $E$ of the neuronal population instead of the spike train of $\delta$-spikes as in the spiking model. Another consequence of this population averaging is that the intrinsic noise of spiking networks is averaged out. To simulate the presence of

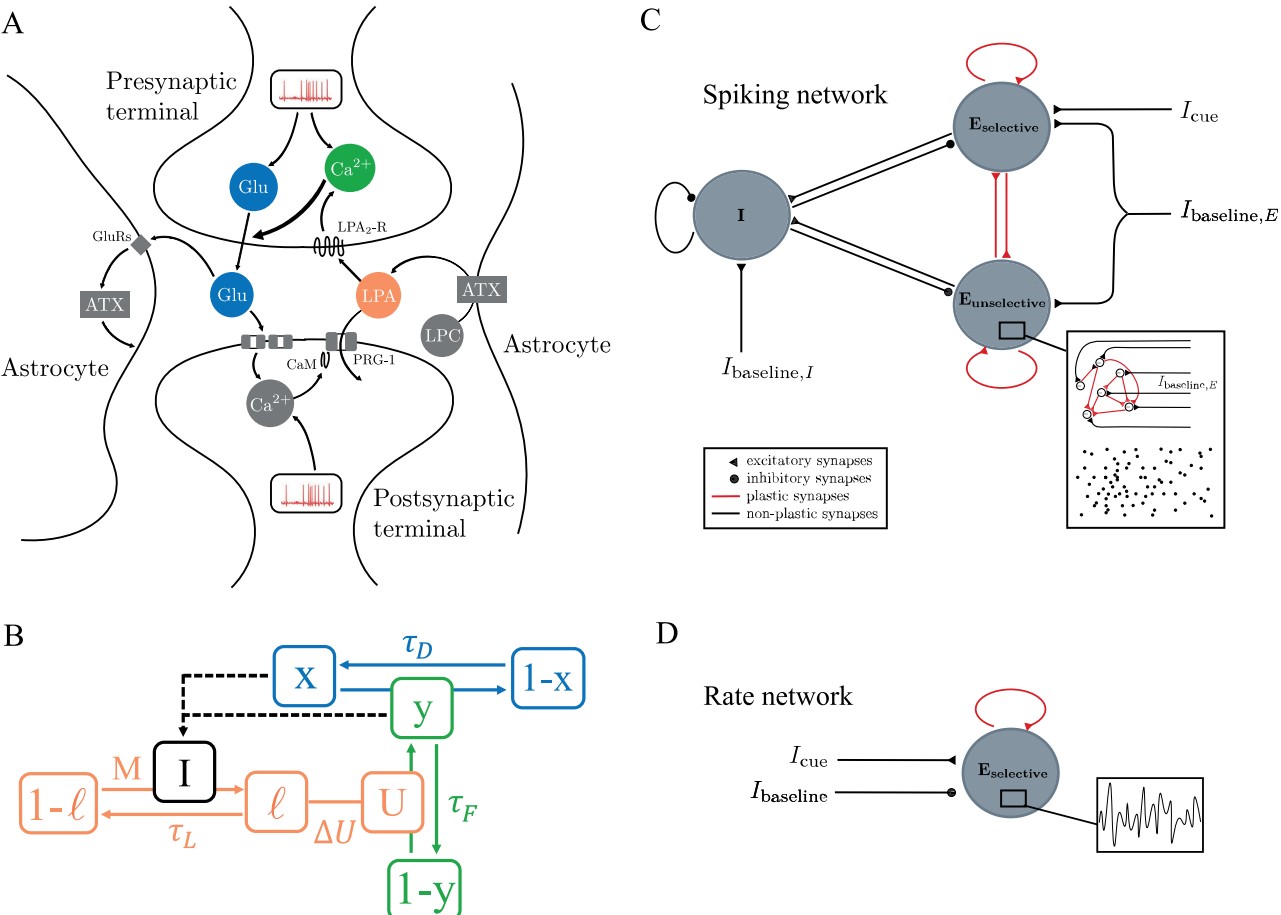

**Fig 1. Synaptic working memory (WM) with astrocytic signaling: Synapse and network model. A** Sketch of the transmission dynamics at tripartite glutamatergic synapses, as determined by the components of presynaptic STP, and its modulation by the astrocytic enzyme ATX [43] and the postsynaptic protein PRG1 [42] via lysophosphatidic acid (LPA). Presynaptic STP influences synaptic transmission via glutamate availability (blue) and calcium-binding (green) at the presynaptic terminal. The astrocytic and postsynaptic components modulate presynaptic calcium-binding rates by increasing the concentration of LPA in the synaptic cleft, which allows more LPA in the synaptic cleft (orange) to bind to the presynaptic LPA2 receptors. Postsynaptic calcium increases LPA concentrations in the cleft by inhibiting the postsynaptic protein PRG1 (via CaM) that usually takes up newly synthesized LPA from the cleft. Astrocytic, ATX-mediated synthesis of LPA is amplified in a activity-dependent way, since some of the synaptically transmitted glutamate binds to astrocytic receptors that increase ATX activity. Implications of elevated calcium-binding due to LPA signaling for paired-pulse-ratio are shown in Fig B in S1 Text. **B** Sketch of the synapse model that captures the transmission dynamics shown in (A). The dynamics of the synaptic variables $x$ (neurotransmitter availability), $y$ (presynaptic calcium binding), and $\ell$ (presynaptic LPA binding) are determined by the STP parameters $U$, $\tau_D$ and $\tau_F$ and the additional astrocytic signaling parameters $\Delta U$ (maximal LPA-mediated increase of $U$), $M$ (activity-dependent increase of $\ell$) and $\tau_L$ (decay timescale of $\ell$). The equations of the synapse model (Eqs 1, 2, 4 and 5) and the dynamics of each variable are described in Models and methods. **C** Sketch of the spiking network model of synaptic WM (architecture as in Mongillo et al. [5]). The network consists of two excitatory populations and one global inhibitory population. Each population consists of sparsely and randomly connected LIF neurons. During the WM protocol, all neurons receive an excitatory baseline input with Gaussian white noise. When the WM cue is presented, the neurons of the excitatory selective population receive an additional external input $I_{cue}$. All except excitatory-to-excitatory (E-E) connections have fixed synaptic weights. All E-E connections underlie either only STP (STP networks) or STP with astrocytic modulation by the LPA-mechanism as sketched in A and B (STP+LPA spiking network). **D** Sketch of the rate network model of synaptic WM. The rate network consists of one excitatory population that is described by a single population firing rate. The recurrent excitatory connections underlie either only STP or STP with astrocytic modulation (LPA-mechanism) as sketched in A and B. We refer to these networks as STP rate networks and STP+LPA rate networks, respectively. During the entire WM protocol, the excitatory population receives a constant baseline input current with Gaussian white noise. This input is inhibitory since it represents the combined effect of global inhibitory and inputs from surrounding (non-selective) excitatory populations. During the cue presentation, it receives an additional excitatory input current $I_{cue}$.

intrinsic noise in the rate network, we therefore introduce Gaussian white noise perturbations of the synaptic variables $x$, $y$ and/or the population activity $E$.

## Astrocyte-modulated STP synapse model

As illustrated in Fig 1A, the LPA signaling mechanism influences STP dynamics by modulating the rate at which presynaptic calcium binds upon spike arrival [42–44]. This effect is mediated by a temporary increase in the concentration of LPA in the synaptic cleft that allows LPA to bind to its presynaptic LPA2 receptors. Upon LPA binding, the G-protein coupled LPA2 receptors induce slow but persistent calcium transients [51] which increase calcium binding to presynaptic active zones and, thereby, lead to more glutamate being released in response to a presynaptic spike. The LPA concentration in the synaptic cleft is modulated by postsynaptic and astrocytic signaling in an activity-dependent way. When a spike is transmitted, the glutamate released into the synaptic cleft binds to receptors in the astrocyte and postsynaptic terminal [44]. The former boosts the synthesis of new LPA molecules by increasing the activity of the astrocytic enzyme ATX which synthesizes LPA from its precursor lysophosphatidylcholine (LPC) [52]. The glutamate that binds to postsynaptic receptors inhibits the activity of the postsynaptic trans-membrane protein PRG1 via calmodulin (CaM) signaling [53]. PRG1 usually takes up LPA from the cleft into the postsynaptic cell [42]. Due to the inhibition of PRG1 activity upon CaM binding, more LPA molecules remain in the cleft and bind to LPA2 receptors in the presynaptic terminal, where they affect presynaptic calcium dynamics as described above. The interplay of the LPA synthesis in astrocytes, the inhibition of PRG1-mediated postsynaptic LPA uptake, and the effects of LPA binding on presynaptic calcium dynamics constitute the activity-dependent LPA signaling mechanism. In contrast to classic synaptic STP, which affects synaptic dynamics in the millisecond- to second-range [21], the LPA-mediated effects of astrocytic signaling in the presynapse occur on longer timescales of several seconds to a minute [51] (see section 'Biological evidence for slow LPA timescales' in S1 Text for details).

Inspired by the approach by De Pitta et al. [31, 50], we integrate the LPA mechanism into the STP synapse model [18] by adding a variable $\ell$ that represents the amount of LPA bound by the presynaptic LPA2 receptors and takes values between 0 (no LPA bound) and 1 (maximal amount of LPA bound). Its dynamics follow

$$\frac{d\ell}{dt} = -\frac{\ell}{\tau_L} + M(1-\ell)I. \tag{4}$$

The increase of $\ell$ depends linearly on the transmitted synaptic current $I$ defined in Eq 3. $M$ determines how much $\ell$ increases with each unit of transmitted current. The gradual unbinding of LPA from the LPA2 receptors is expressed as decay of $\ell$ with time constant $\tau_L \gg \tau_F, \tau_D$. Bound LPA $\ell$ increases the presynaptic calcium-binding rate $U$:

$$U(t) = U_b + \ell(t)\Delta U, \tag{5}$$

where $U_b$ and $\Delta U$ are fixed parameters corresponding to the baseline calcium-binding rate and the maximal LPA-mediated increase above that baseline, respectively. To stay consistent with the STP model, where $U \in [0, 1]$, the choice of the parameter $\Delta U$ is limited to the interval $[0, 1 - U_b]$ for a given baseline calcium-binding rate $U_b \in [0, 1]$. Combined with the STP equations above, Eqs 4 and 5 capture the modulation of presynaptic STP dynamics and the synaptic release rate by the LPA mechanism. Note that we consider each synapse to be interacting with an independent astrocytic compartment that does not communicate with astrocytes or astrocytic compartments surrounding other synapses, i.e. each synapse in our model operates using its own astrocytic variable $\ell$. Future generalisations of the model could consider spatial

astrocyte-to-astrocyte interactions [54] as diffusive coupling of astrocytic intracellular IP3 and calcium concentrations in a lattice of partly overlapping astrocytes, effectively correlating $\ell$ variables of different synapses. Astrocyte-modulated STP synapses in rate networks can be analogously defined by extending the rate formulation of the STP synapse model with an equation for the population-averaged variable $\ell$ (see Eqs 9, 10 in section 'Firing rate network: model and parameters' in S1 Text). Fig 1B visualizes the relationship between STP and astrocytic variables and parameters as described above.

## Network models

We compare two models of synaptic WM: one with and one without astrocytic modulation of STP. To allow for different methods of analysis, we implement each as a rate and a spiking network model, respectively (Fig 1C and 1D). For the synaptic WM model without astrocytic modulation, we re-implemented the synaptic attractor spiking and rate model introduced in the seminal work of Mongillo et. al [5]. In the following, we will refer to them as STP spiking and STP rate network, respectively. To model the influence of the astrocytic LPA mechanism on WM, we integrate our STP synapse model with LPA signaling as described in the previous section (Eqs 1, 2, 4 and 5) in the synaptic STP spiking and rate models by Mongillo et. al [5]. We will refer to them as STP+LPA spiking and rate networks, respectively. See Fig 1C and 1D for illustrations and detailed description of spiking and rate network architectures. A full model description is also provided in sections 'Spiking network: model and parameters' and 'Firing rate network: model and parameters' in S1 Text and in Tables A and C in S1 Text.

## Results

### LPA signaling modulates synaptic WM durations and introduces bimodal distribution with short transient and persistent representations

Our hypothesis is that activity-dependent astrocytic modulation of synaptic STP dynamics can influence WM representations. To test it, we compare the network dynamics arising during the delay period of the WM protocol in rate and spiking networks without and with astrocytic signaling (STP and STP+LPA rate and spiking networks, see Models and methods). As we vary the synaptic or external input parameters, we observe that all four network types exhibit the same three WM representations (see Fig 2A for a representative example from an STP +LPA spiking network, equivalent dynamics for other networks are shown in subsequent figures). After the presentation of the WM cue, the excitatory selective population either (i) returns to asynchronous firing at the same rate as before the cue (silent regime, [5]), (ii) shows a finite number of population spikes before returning to pre-cue asynchronous firing (transient regime), or (iii) enters sustained synchronized population spiking (persistent regime, [5]), depending on the network parameters. The silent regime (i) corresponds to a stable silent state (fixed point), whereas the persistent regime (iii) corresponds to a stable active state (limit cycle). The transient regime corresponds to a meta-stable active state, i.e. an oscillatory state that is sufficiently unstable to be stochastically terminated by noise, which coexists with a stable silent state, i.e. a fixed point that is sufficiently stable to prohibit noise-induced reappearance of the meta-stable active state. Additionally, the networks can also exhibit chaotic [28] or asynchronous activity [55]. Throughout our work, we will focus on the transient WM regime (center black box in Fig 2A) and its transitions to the silent and persistent WM regime. The parameter ranges for the different regimes depend on whether astrocytic signaling is present or not, suggesting that astrocytic modulation could shift a network that is usually in a silent WM regime to a transient or persistent WM regime or vice versa. Even more importantly,

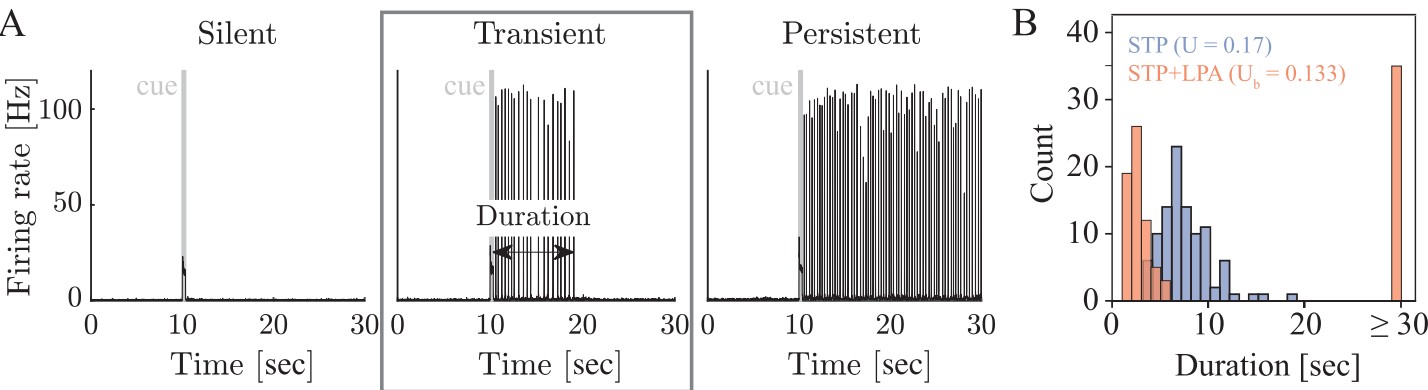

**Fig 2. Astrocytic signaling leads to differences in synaptic WM termination. A** Panels show the firing rate of the excitatory selective population of a representative STP +LPA spiking network during the WM protocol for three different levels of external baseline input current (remaining network parameters as in Table B in S1 Text). The network responds to the presentation of the WM cue (grey bar at 10s) with a silent (left, $\mu_E\tau_E = 22.95$ mV), transient (center, $\mu_E\tau_E = 23.165$ mV) or persistent (right, $\mu_E\tau_E$ = 23.4 mV) WM representation. Equivalent WM regimes are observed in STP+LPA rate networks and STP spiking and rate networks (see subsequent figures). **B** Distribution of durations of the WM representations in STP and STP+LPA spiking networks in the transient regime for a fixed set of parameters (see subsequent figures). Even though both networks produce transient WM representations, the duration of the WM representation follows a homogeneous distribution in the STP network, but a bimodal distribution in the STP+LPA network. The same observations can be made for STP and STP+LPA rate networks (shown in subsequent figures).

however, we can observe a qualitative difference in the termination of transient WM representations with and without astrocytic modulation (Fig 2B): While the stochastic durations of transient WM representations in STP networks are typically homogeneously distributed, they can follow a bimodal distribution in STP+LPA networks. This observation holds for both spiking (Fig 2B) and rate networks (shown in subsequent figures) and confirms our hypothesis that astrocytic modulation of STP via the LPA-mechanism can affect WM representations. It also raises the question how this effect arises mechanistically from the network dynamics. As a first step towards an answer, we need to understand the mechanisms by which the active WM representation terminates in the transient regime. To this end, we take two steps back and look at the phase space dynamics of STP rate networks with noise.

## Noise can turn activity-silent and persistent firing regimes into transient WM activity

To identify the network mechanisms by which noise can lead to the stochastic termination of WM representations, we consider STP rate networks (see Fig 1D) with three different sources of noise: (i) noise in the amount of presynaptically bound calcium $y$ (Eq. 6 in S1 Text), (ii) noise in the amount of available neurotransmitter vesicles $x$ (Eq. 7 in S1 Text), and (iii) noise in the network activity $E$ (Eq. 8 in S1 Text). The assumption of noise in the synaptic variables $x$ and $y$ is biophysically motivated by the intrinsic stochasticity of the molecular processes by which calcium binds to the presynaptic active sites and neurotransmitter vesicles are released, recycled and refilled in biological synapses. The noise in the network activity $E$ reflects the 'network noise' that arises in biological (and spiking networks) from the stochastic spike generation process, distributed synaptic delays and other sources of randomness in the network interactions.

For simplicity, consider a STP rate network with fixed synaptic parameters $\tau_D$, $\tau_F$, and $U$. Its bifurcation diagram (Fig 3A, see Refs. [5, 24–28] for an in-depth analysis) shows that the network dynamics traverse the following regimes as the baseline input increases: It first exhibits (i) a stable fixed point ($I_{base} < -3.25$) with low population activity, then (ii) a bistable regime ($-3.25 < I_{base} < -2.5$) with a low-activity fixed point (down-state) and a limit cycle (up-state), where the cue stimulus can switch the system from the down- to the up-state, (iii) a stable limit

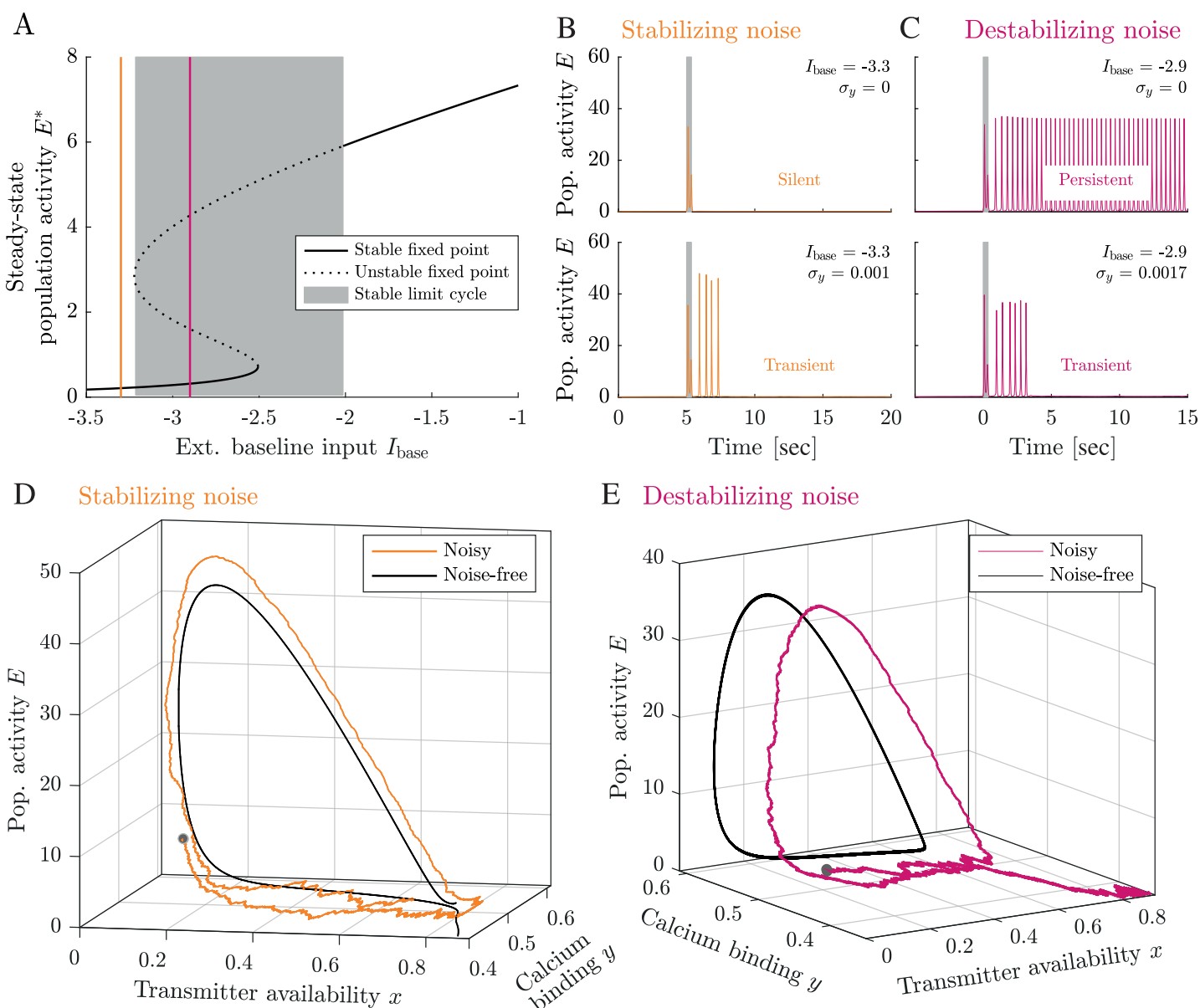

**Fig 3. Noise can turn activity-silent and persistent firing regimes into transient WM activity. A** Bifurcation diagram for a STP rate network without LPA signaling. The grey-shaded area denotes the regime in which a stable limit cycle exists. Presenting a sufficiently strong WM cue to silent networks in the bistable regime evokes persistent WM activity. **B-C** Population activity for two rate network simulations in the silent and the persistent regime, with and without noise (of amplitude $\sigma_y$) added to the synaptic variable $y$. The grey shaded area denotes the cue stimulus. In the silent regime ($E < -3.25$), noise can stabilize a 'ghost' of a limit cycle [56] ('stabilizing noise'). In the regime of persistent limit cycles, noise can disrupt oscillations ('destabilizing noise'). **D-E** Typical trajectories of transient limit cycles emerging from stabilizing (orange line, D) and destabilizing noise effects (magenta line,E) at baseline input levels $I_{base}$ = -3.3 (left) and $I_{base}$ = -2.9 (right). Black lines show a representative example of a noise-free trajectory (left) and the limit cycle (right). Remaining rate network parameters: see Table D in S1 Text.

cycle ($-2.5 < I_{base} < -2$) and, finally, (iv) a stable, high-activity fixed point ($I_{base} > -2$). When we add noise to the system, we observe that transient oscillations emerges in two regions of the phase space: (i) in the region of the stable down-state, and (ii) in the bistability region. Simulations in the phase space of the STP rate network show that these transient oscillations can be induced by noise in two ways. In the region of the stable down-state, noise transiently 'stabilizes' an unstable cycle in the phase space (Fig 3B), while in the bistability region, noise

'destabilizes' the existing stable limit cycle (Fig 3C). In the following, we discuss the two mechanisms in detail.

To understand how 'stabilizing' noise effects can produce transient WM activity (Fig 3B), we consider a noise-free STP rate network in the stable fixed point regime. Before stimulation with the WM cue, the network is in a steady state at its low-$y$, high-$x$, low-$E$ fixed point. When the WM cue is presented, a stable limit cycle forms in the phase space. The increased external input due to the WM cue moves the dynamics of $x$, $y$ and $E$ into this limit cycle region, where the trajectory describes one (or several) population spikes until the cue stimulus ends. After cue offset, the limit cycle looses its stability and the trajectory returns towards its stable low-rate fixed point. However, in the location of the previously stable limit cycle, an unstable rotational component remains in the phase space ('ghost' of the limit cycle, black line in Fig 3D). While returning to the stable fixed point, the trajectory of $x$, $y$ and $E$ passes close to this 'ghost' of the limit cycle, at the high-$y$, high-$x$, low-$E$ state from which the population spikes originated during cue presentation. Whereas the noise-free system returns to the fixed point from there, the trajectory is so close to the 'ghost' of the limit cycle that, in a system with noise, the noise can push the network into another cycle, inducing another population spike (orange line in Fig 3D). Each noise-induced population spike makes the system pass close to the high-$y$, high-$x$, low-$E$ from which a population spike can still originate. This facilitates re-occurring noise-driven population spikes, even in the absence of a stable limit cycle—the noise 'stabilizes' the transient oscillation and thereby leads to a stochastic number of population spikes after the WM cue stimulus in STP rate networks with noise. For a system without facilitation (i.e. only $E$ and $x$ variable), small deterministic fluctuations can cause a fast rotation around the 'ghost' of the limit cycle [56] and produce a similar meta-stable limit cycle. In contrast to the case we consider here, however, they are driven by the deterministic dynamics of the network [24–26, 57].

The second way in which the transient WM regime appears in STP rate networks with noise is through noise-induced destabilization of a limit cycle. This case of transient WM arises in a parameter regime where the noise-free STP network would exhibit persistent activity, i.e. a stable limit cycle (Fig 3C). In this case, the network dynamics are governed by two stable attractors: a low-activity fixed point and a limit cycle (black line in Fig 3E). Sufficiently strong noise can destabilize the limit cycle (purple line in Fig 3E). The destabilization takes place during the low-activity phase of the cycle when the system is close to the separatrix of the two attractors, i.e. the boundary separating the regions of phase space with fixed point and limit cycle dynamics.

The impact of stabilizing and destabilizing noise effects depends on the nonlinear dynamics of the system close to the separatrix (see Figs F—H in S1 Text). For example, the STP rate network dynamics are nonlinear around the separatrix in the sense that positive fluctuations along the dimension of the fast variable $E$ have very large effects, while negative fluctuations along $E$ have very small effects. This nonlinear behaviour (also along the $y$ and $x$ dimensions) is reflected by the length of the flow arrows in the phase plots (Fig J in S1 Text). The nonlinear dynamics of the network activity $E$ close to the separatrix explain why adding noise to the population activity $E$ tends to stabilize transient WM activity (Fig F in S1 Text). Negative fluctuations along $E$, which would prevent new population spikes by destabilizing the limit cycle in the bistability regime, have very small effect on the dynamics trajectory. In contrast, already small positive fluctuations of $E$ in the fixed point regime can induce a cycle in the phase space (i.e. a population spike in the network activity), following the 'ghost' of the limit cycle (see Fig J in S1 Text and Refs. [25, 56]). In contrast, adding noise to the synaptic variable $y$ tends to disrupt transient WM activity (Fig H in S1 Text). This is because a noise-induced decrease of $y$ can push the fast $E$-$x$-system to cross the separatrix towards the fixed point and cause a

subsequent collapse of oscillatory activity (see Fig J in S1 Text). This explains why noise can have predominantly stabilizing or destabilizing effects pending on the source of the noise, i.e. whether we assume noise in $x$, $y$ or $E$.

Note that here, we only consider regimes in which the noise on any of the three variables is not strong enough to induce transient population spikes without the prior presentation of the WM cue. Higher noise levels, which can lead to spontaneous re-appearance of transient WM representations, could be a promising model of free (i.e. spontaneous, not cued) recall of WM items. Since we focus on the spontaneous termination of active WM representations, we exclude the corresponding high-noise regimes for the remainder of this study.

## Effects of STP parameters on the mean transient WM duration

In the last section, we have shown how noise can lead to transient WM representations. However, noise alone does not explain why STP networks with and without LPA signaling show qualitatively different distributions of transient durations (Fig 2B). In the following sections, we will shed light on the mechanisms underlying this phenomenon. The influence of LPA signaling on STP dynamics is mediated by its influence on the calcium binding rate $U$ (Eqs 4 and 5). Therefore, we begin by investigating how the calcium binding rate $U$ (and other STP parameters) influence the distribution of transient WM durations in STP rate and spiking networks.

Fig 4A and 4G show the distributions of durations of active WM representations for changing calcium binding rate $U$ in the STP rate network with noise and the STP spiking network, respectively. As $U$ increases, both rate and spiking networks shift from silent WM representations ('zero duration') to transient WM representations of continuously increasing duration (Fig 4A and 4G). When the duration of transient representations exceeds the observation window, we refer to them as persistent WM representations. For each value of $U$, the distribution of transient durations is unimodal around its mean, with a longer tail towards higher durations (insets in Fig 4A and 4G). The mean of the distribution increases continuously with the calcium binding rate $U$, whereas the shape of the distribution does not change significantly.

Fig 4B, 4C, 4H and 4I show the population activity and synaptic variables during two transient WM representations with different duration that emerge for the same value of $U$ in the STP rate and spiking network, respectively. In both cases, the presentation of the WM cue pushes the system from a low-$E$, high-$x$, low-$y$ steady state into a regime where the network cycles between high and low $E$ values (population spikes). The values of $x$ and $y$ oscillate around overall lower (for $x$) and higher (for $y$) levels than before the cue presentation. After an initial adaptation period, the values of $x$ and $y$ settle around a stable level, from which noise stochastically disrupts the cycle by pushing the network dynamics back into the pre-cue low-$E$, high-$x$, low-$y$ regime.

The bifurcation diagram of the the underlying noise-free STP rate network (Fig 4D) shows that for a sufficiently strong cue stimulus, the transients appear around the bifurcation between (i) fixed point regime and (ii) bistable region along $U$ (highlighted in grey). In this parameter region, the cue stimulus is strong enough to induce population spikes. After the cue offset, noise leads to a long-tailed distribution of transient durations via stabilizing or destabilizing noise effects, as described in the previous section. If the cue stimulus is weak enough, then the transition is postponed to higher values of $U$ (Fig I in S1 Text). Projecting the trajectories onto $y$-$x$-space (Fig 4E and 4F) shows why the mean of the transient representation increases continuously. As $U$ is increased, a stable limit cycle emerges, and, subsequently, the distance of the dynamics trajectory to the separatrix increases. These two effects increase the stability of the transient oscillation against disruption by noise and thereby lead to longer average transient representations.

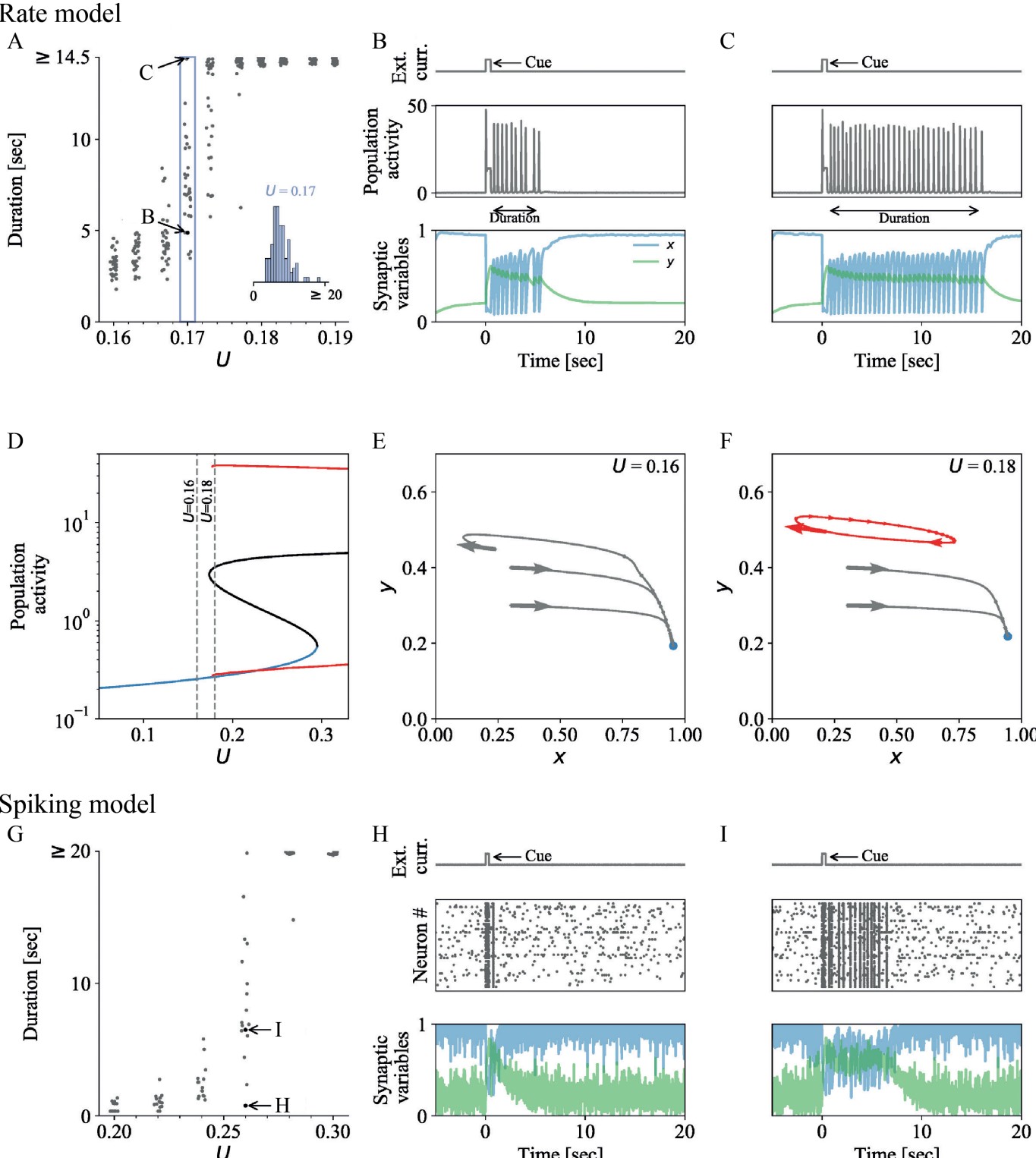

**Fig 4. Transient WM activity is modulated by synaptic dynamics in STP rate and spiking networks without LPA signaling (ΔU = 0). A-F** Analysis of the STP rate model (see Models and methods, Table C in S1 Text; network parameters as in Table D in S1 Text except otherwise specified). **A** WM durations for 30 simulations at 10 values of $U_b$. As the phase transition is crossed, the range of durations of transient population activity upon cue stimulation increases continuously (a hysteresis effect emerges if the cue is weak, see Fig I in S1 Text). Inset: Distribution of transient WM durations of 100 simulations at $U_b = 0.17$. **B-C** Two simulations at $U_b = 0.17$. The external input current with the cue stimulus is shown in the top panel, population activity in the center and synaptic traces

(vesicle availability $x$, Calcium concentration $y$, and LPA binding $\ell$) are shown in the bottom panel. B and C are both in a metastable regime; after 5 and 15 seconds, respectively, the synchronized population activity that emerges after cue presentation is disrupted and the population activity and the synaptic variables return to their baseline values. **D** Bifurcation diagram, showing stable (blue) and unstable (black) fixed points, as well as the maximal and minimal amplitude of stable limit cycles (red solid line). **E-F** Analysis of the system before and after the phase transition for $U_b = 0.16$ and $0.18$, respectively. Projections of three trajectories of a system without noise from initial conditions $(x, y) = \{(0.3, 0.3), (0.3, 0.4)\}$, which return to the fixed point, and (for E) the last cycle for a simulation with initial condition $(x, y) = (0.5, 0.5)$, onto the $x$-$y$-plane. At the initial state, $E$ is set to be at the lower equilibrium state ($dE/dt = 0$) for the given $x$, $y$. In F, the limit cycle is shown in red. **G-I** Analysis of a STP spiking network model (see Table A in S1 Text). Parameters as in Table B in S1 Text except otherwise specified. **G** Transient WM durations for 15 simulations for 6 values of $U_b$. As $U_b$ increases, the WM cue is first followed by a silent synaptic trace, then by transient and persistent population spiking. Within a small window of values, $U$ modulates the median duration of the transient, actively spiking WM representation. **H-I** Two simulations for $U_b = 0.26$ with different transient durations. Panels as in B and C but the subplot in the center shows a rasterplot of 50 neurons of the selective population instead.

## Weak LPA signaling implements slow modulation of the mean transient WM duration via its influence on presynaptic calcium binding

Now we return to rate and spiking networks with LPA signaling. In these networks, the STP dynamics are influenced by the dynamics of the LPA binding variable $\ell$ through its impact on the calcium binding probability $U$ (Eq 5). The $\ell$-modulated dynamics of $U$ in the networks with LPA signaling depend on three parameters: the LPA binding rate $M$, the (slow) timescale of the LPA unbinding $\tau_L$, and the LPA-mediated increase of the presynaptic calcium binding probability $\Delta U$. First, we consider the impact of the slow astrocytic timescale $\tau_L$ on the distribution of transient durations, for the scenario of weak astrocytic modulation (low $\Delta U$).

Fig 5A and 5D show the distributions of transient WM durations for changing $\tau_L$ and weak astrocytic signaling effects (low $\Delta U$) in STP+LPA rate networks with noise and STP+LPA spiking networks, respectively. Like in STP networks, the distribution of transient durations is unimodal with a long tail (Fig 5A and 5D). However, the consistent increase of mean and median duration with $\tau_L$ is mainly due to an elongation of the tail at high durations instead of a homogeneous shift of the distribution as a whole, like in STP networks. In networks with weak LPA signaling, longer transient WM representations therefore come with a higher variability of the durations. Since increasing $\tau_L$ effectively increases the calcium binding rate $U$ via the steady-state values of $\ell$ (Eqs 4 and 5), increasing the value of $\tau_L$ moves the system dynamics right in the bifurcation diagram, i.e. in the direction where the limit cycle gains stability. Hence, increasing $\tau_L$ shifts the limit cycle away from the separatrix, such that it becomes less susceptible to noise disruption—as a result, the distribution of WM durations becomes wider.

The plots of the synaptic dynamics during the transient WM activity (Fig 5B, 5C, 5E and 5F) show that, in the beginning of the transient activity, $x$ and $y$ show similar behavior as in STP networks—they increase ($y$) or decrease ($x$) during cue presentation and stabilize in an oscillatory state after the cue offset before being disrupted by noise. After the offset of the cue, $\ell$ ramps up while the active WM representation persists (Fig 5B, 5C, 5E and 5F). Due to its dependence on the synaptically transmitted current ($xy$, see Eq 4) and the slow timescale of its decay, $\ell$ 'integrates' the population spikes that occur during the WM. However, since the influence of astrocytic signaling on the synaptic efficacy is small in this case (low $\Delta U$), the ramping dynamics of $\ell$ do not significantly influence the dynamics of the presynaptic variables (Fig 5B, 5C, 5E and 5F).

## Strong astrocytic signaling introduces bimodal distribution of transient WM durations and 'window of vulnerability' for WM termination

Next, we consider how the baseline calcium binding rate $U_b$ influences transient WM representations in a scenario with strong astrocytic modulation (high value of LPA-mediated presynaptic effect $\Delta U$).

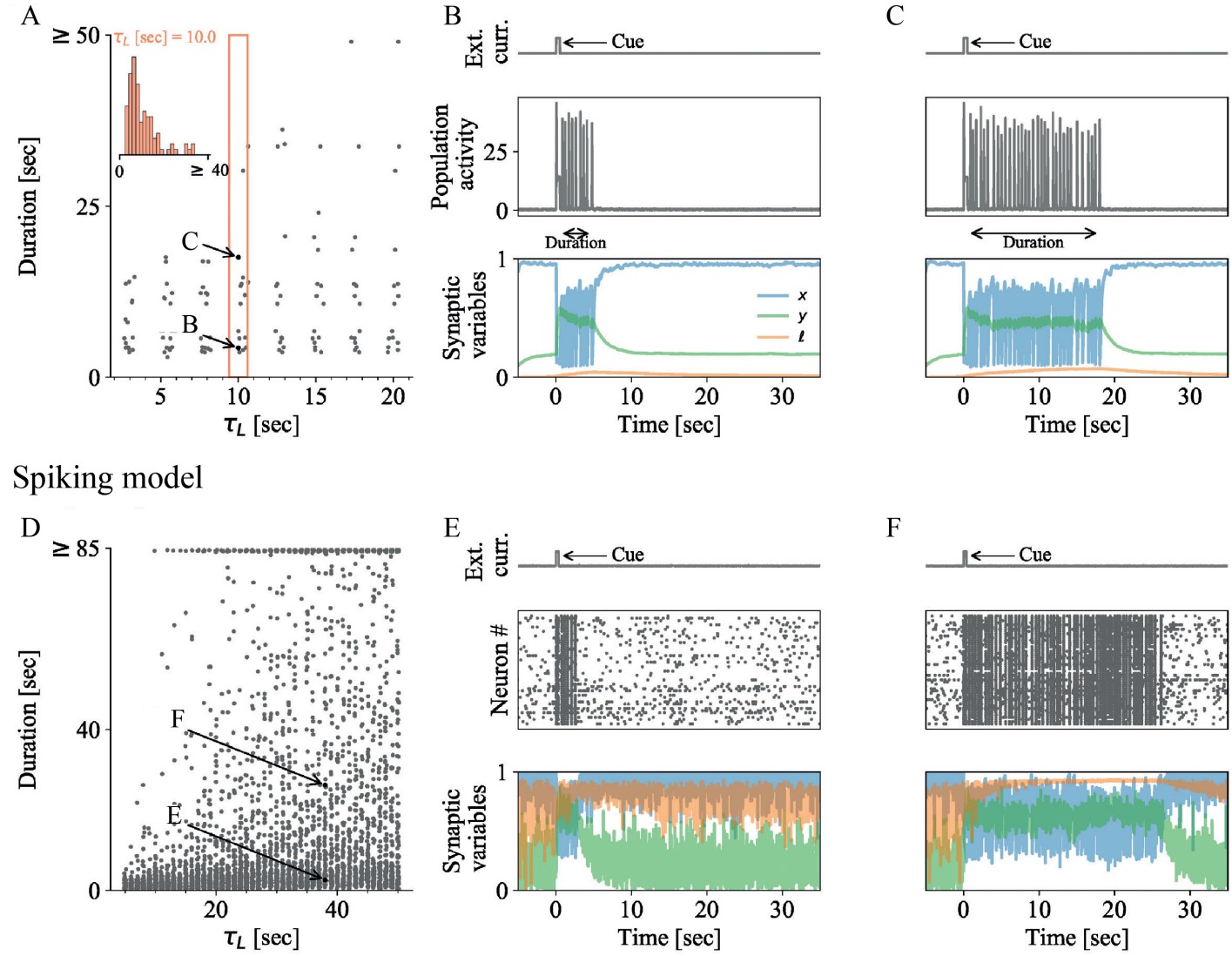

**Fig 5. Weak LPA signaling implements slow modulation of the mean transient WM duration via its influence on presynaptic calcium binding. A-C** Analysis of the STP+LPA rate model (see Models and methods and Table C in S1 Text). Parameters: $\Delta U = 0.1$, $M = 0.2$, $U_b = 0.16$, $\sigma = 0.1$, remaining parameters: Table D in S1 Text. **A** WM durations for 15 simulations at each of eight values of $\tau_L$. Durations of transient activity slowly increase with $\tau_L$. Inset: Distribution of transient WM durations of 100 simulations at $\tau_L = 10$sec. **B-C** Two simulations at $\tau_L = 10$sec. Panels as in Fig 4. **D-F** Analysis of a STP+LPA spiking network model (see Models and methods, Table A in S1 Text). Parameters: $\Delta U = 0.1$, $M = 0.6$, $U_b = 0.2$, remaining parameters: Table B in S1 Text. **D** Transient WM durations for 100 simulations for each of 46 values of $\tau_L$. The time constant of presynaptic LPA unbinding $\tau_L$ regulates the duration of transient active WM representations across a broad range of biologically realistic values. **E-F** Two simulations for $\tau_L = 38$sec with different transient durations. Panels as in B and C but the subplot in the center shows a rasterplot of 50 neurons of the selective population instead.

Fig 6A and 6G show the distribution of transient WM durations for changing baseline calcium binding rate $U_b$ and strong astrocytic modulation in STP+LPA rate networks with noise and STP+LPA spiking networks, respectively. For very low baseline calcium binding rates ($U_b \leq 0.165$), the durations are homogeneously distributed around the mean and similar to the distribution of transient durations in the STP network (Fig 4A and 4G). As $U_b$ increases, however, the distribution becomes bimodal with a homogeneously distributed bump at low durations and a concentrated peak at the maximal observable duration. In this regime, the WM

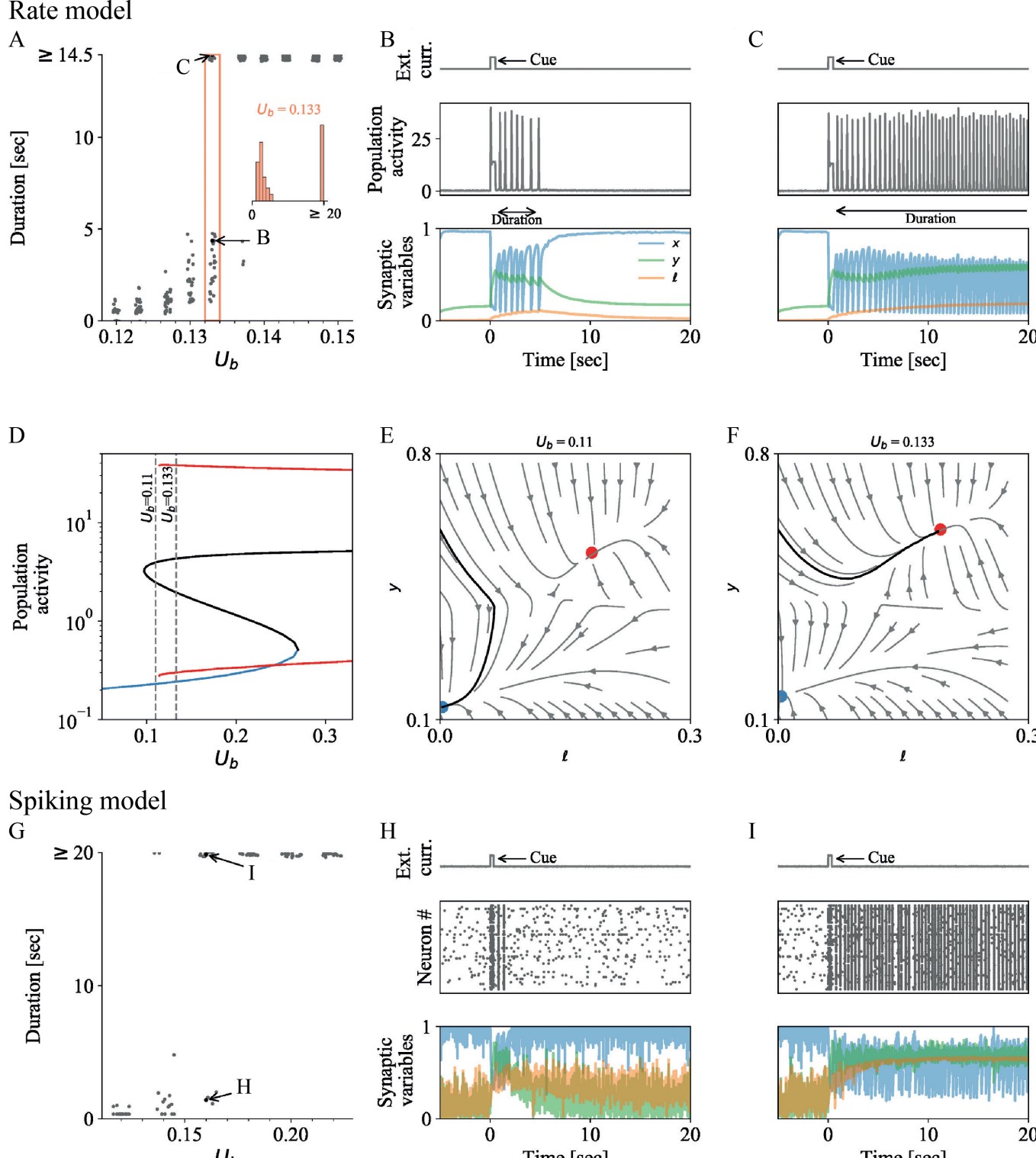

**Fig 6. Strong astrocytic signaling introduces bimodal distribution of transient WM durations and 'window of vulnerability' for WM termination. A**
Analysis of the STP+LPA rate model (see Models and methods, Table C in S1 Text). Parameters: $M = 0.4$, $\Delta U = 0.4$, $\tau_L = 5$sec, remaining parameters: Table D in
S1 Text. **A** WM durations for 30 simulations at each of 10 values of $U_b$. As the phase transition is crossed, the range of durations of transient population activity
upon cue stimulation increases continuously (a hysteresis effect emerges if the cue is weak, see Fig I in S1 Text). Inset: Distribution of transient WM durations of
100 simulations at $U_b = 0.133$. **B-C** Two simulations at $U_b = 0.133$. Panels as in Fig 4. **D** Bifurcation diagram, showing stable (blue) and unstable (black) fixed

points, as well as the maximal and minimal amplitude of stable limit cycles (red solid line). **E-F** Direction field of the two slow variables $y$ and $\ell$ within $[0, 0.3] \times [0.1, 0.8]$, if a separation of time scales to the faster variables $E$ and $x$ is assumed (see Models and methods for details). The blue and red dot denote the stable fixed point and stable limit cycle, respectively. The black line is a trajectory for a system without noise with initial condition $(y, l) = (0.6, 0)$. See Fig K in S1 Text for a comparison of the scale-separated slow variables and the full system for $U_b = 0.133$. Distribution of transient durations for different noise levels and noise sources are shown in Figs F-H in S1 Text. Here, $\sigma_E = 0.05$, $\sigma_{x,y} = 0$. **G-I** Analysis of a STP+LPA spiking network model (see Table A in S1 Text). Parameters: $M = 0.4$, $\Delta U = 0.25$, $\tau_L = 9$sec, remaining parameters: Table B in S1 Text. **G** Transient WM durations for 15 simulations for each of 6 values of $U_b$. As $U_b$ increases, the WM cue is first followed by a silent synaptic trace, then by transient and persistent population spiking. Within a small window of values, $U$ modulates the median duration of the transient, actively spiking WM representation. **H-I** Two simulations for $U_b = 0.16$ with different transient durations. Panels as in B and C but the subplot in the center shows a rasterplot of 50 neurons of the selective population instead.

representations are either transient but short ($< 10$ seconds, low-duration bump), or persistent, i.e. not disrupted within the simulation time (high-duration peak). As $U_b$ increases further, the low-duration bump maintains its shape and slowly shifts upwards, similar to the STP case (Fig 4A and 4G). The high-duration peak, on the other hand, grows larger with $U_b$, until the two components of the distribution become indistinguishable for very high calcium binding rates ($U_b \geq 0.18$), due to the final observation time. The bifurcation diagram (Fig 6D) shows that with strong astrocytic modulation, the transient regime appears within the region where the limit cycle is already stable.

The synaptic dynamics of the underlying noise-free rate network (Fig 6E and 6F) show that the low-duration bump can be explained by the synaptic dynamics of the slow variable $\ell$. In particular, the slow ramping in combination with the long timescale of $\ell$ introduces a 'window of vulnerability' at short durations, where transients are more susceptible to disruption by noise, before gaining stability at longer durations. For the phase space analysis of the synaptic dynamics, we separate the fast dynamics of population activity $E$ and depression variable $x$ from the slower variables $y$ and $\ell$ (see Models and methods) as in the previous section. The trajectories of the slow variables reveal that after the offset of the cue stimulus, $y$ decreases and starts approaching the separatrix between limit cycle (where $y$ is higher) and fixed point (where $y$ is lower). In this brief regime of low $y$ values, noise can push the trajectory over the separatrix into the fixed point region, which results in the termination of the transient WM activity. Meanwhile, $\ell$ slowly ramps up due to the repeated population spikes, and causes $y$ to increase again after the initial low-$y$ regime. This pushes the trajectory of synaptic dynamics away from the fixed point-region and stabilizes the transient representation. This stabilizing effect of $\ell$ is so strong that the WM representations that survive the initial low-$y$ time window persist until the end of the observation window. We also refer to the low-$y$ phase as 'window of vulnerability' during which noise can disrupt a transient WM representation and after which the representation stabilizes.

This mechanism is reflected in the traces of the synaptic variables over time (Fig 6B, 6C, 6H and 6I). Calcium binding $y$ decreases immediately after the offset of the cue, destabilizing the oscillations. Here, noise can easily disrupt the transient activity (Fig 6B and 6C). Synaptic LPA $\ell$ increases gradually after the offset of the cue stimulus (orange traces in Fig 6B, 6C, 6H and 6I) as long as the memory is active, and pushes the calcium binding $y$ back up (after 5 seconds after cue stimulus in Fig 6C and 6I). This stabilizes the transient WM activity and makes it more robust against disruption by noise. Figs M—O in S1 Text show that, indeed, $y$ is lower and $x$ is higher in the two last population spikes before termination of transient WM representations in the respective spiking networks with astrocytic signaling.

## Discussion

We studied the duration of transient WM activity in a network model with and without astrocytic modulation of short-term plasticity (STP), and found that astrocytic modulation can

influence WM duration. Transient WM activity is a finite period of population spiking which spontaneously terminates and is followed by a silent synaptic trace. Its termination does not require external stimuli, voluntary reorientation of attention to other WM items, or distractors [58, 59] and could therefore represent the involuntary 'forgetting' of memory items. The astrocytic pathway we consider in our model involves the lipid messenger molecule lysophosphatidic acid (LPA) [44] and modulates short-term synaptic plasticity (STP) on timescales of several seconds to a minute [51], introducing a longer timescale to the fast dynamics of STP [21] (see Fig 1A). Functionally, the signaling pathway could be central for sensory gating [44, 60] and other higher cognitive functions. Our findings shed light on three main aspects: (i) the role of noise and different noise sources in stabilizing or terminating WM activity; (ii) the impact of synaptic parameters and astrocytic modulation on the kind of WM representation that evolves and its stability; and (iii) the effect of the slow astrocytic timescale on the patterns of WM termination.

We show that noise helps generate transient WM activity at the transition between the silent and the persistent WM regime and can explain the stochasticity of transient WM durations around a mean (Fig 2B blue histogram). Importantly, 'noise' in the context of our work refers to the fluctuations of network activity around its mean but does not represent a signal in its own right. We found two mechanisms through which noise can generate stochastic, transient WM durations. In the silent regime, where network dynamics are dominated by a stable low-activity fixed point, sufficiently strong noise can stabilize transient oscillations in the absence of a stable limit cycle. Similar stabilizing effects of noise in neural network dynamics have been studied theoretically [24, 25] and have been reported in the context of long-term memory rehearsal [61] and aperiodic attractors in networks of the olfactory system [62]. In the persistent regime, where the network dynamics exhibit bistability of a low-activity fixed point and a limit cycle, sufficiently strong noise can terminate WM activity by destabilizing the limit cycle. Intrinsic noise in biological as well as rate and spiking networks can have multiple sources, such as noise in the synaptic variables, stochastic spike arrival, and different fixed but random connectivity. Our analysis of the different noise components suggests that in STP firing rate networks, adding noise to the population activity $E$ tends to stabilize transient cycles (see Fig F in S1 Text), whilst noise in the synaptic calcium binding variable $y$ tends to have a disruptive effect (see Fig H in S1 Text). In our STP spiking networks, we identified connectivity noise (also: 'frozen noise' [63]) as an important source of variability that influences the termination of population activity during WM delays (Figs P and Q in S1 Text). These differences in connectivity could originate from differing eigenvalue and rank structures of the connectivity matrix [64, 65] or the prevalence of certain circuit motifs [66]. A detailed analysis of different circuit motives and the impact of resulting spiking correlations on the WM representations lies out of the scope of this work but remains an important direction for follow-up research.

Whilst noise is a important element of transient WM termination, we show that the duration of transient WM representations is highly dependent on presynaptic STP as well as LPA-mediated postsynaptic and astrocytic modulation of STP dynamics. In rate and spiking STP networks without LPA signaling, changes in the presynaptic calcium binding rate lead to a transition between silent (for low calcium binding rate) and persistent (for high calcium binding rate) via transient WM representations of different durations. The higher the calcium binding rate, the longer the duration of the transients. Since the astrocytic mechanism acts on the presynaptic STP dynamics by increasing the presynaptic calcium binding rate, it is a particularly relevant variable for our study. Similar transitions between WM regimes are also possible, e.g. due to changes in the timescales of presynaptic facilitation and depression. A phase space analysis of rate networks with and without astrocytic signaling shows that the influence of synaptic parameters on the duration of transient WM activity stems from the fact that they

determine the regime of the underlying noise-free network dynamics, in particular the existence and stability of a low-activity fixed point (corresponding to baseline network activity) and a limit cycle (corresponding to an active WM representation). In networks without astrocytic signaling, the duration of transient WM representations is directly coupled to the stability of the limit cycle, which in turn is determined by synaptic parameters that are fixed throughout the WM protocol. In contrast, in networks with astrocytic signalling, the slow dynamics of presynaptic LPA binding ($\ell$) allows for a modulation of the stability of WM activity during the course of the WM protocol and even while the WM representation is active. Since the amount of LPA bound to presynaptic receptors increases in response to synaptic transmission, and since the unbinding from these presynaptic receptors is very slow, the amount of presynaptically bound LPA ramps up slowly during the course of the WM activity until its equilibrium value is reached or the activity terminates. We can think of LPA as a slow integrator of recent synaptic activity which, in this case, saves information about the currently active WM representation.

Via its influence on the presynaptic calcium binding rate, LPA then modulates the stability of the faster STP dynamics over the course of the WM maintenance. Depending on the strength of LPA-mediated effects on the presynaptic calcium binding rate (determined by $M$ or $\Delta U$), this can lead to different effects on the distribution of transient WM durations. Weak LPA signaling leads to a widening of the distribution of transient durations since LPA stabilizes the WM activity through integration of the synaptic current—an effect that increases with the timescale of LPA unbinding ($\tau_L$). Strong LPA signaling, on the other hand, leads to a bimodal distribution that consists of short transient and stable persistent WM representations. This effect is due to a 'window of vulnerability', during which network dynamics are first destabilized before they are stabilized by a slow but consistent ramping LPA, and different from the pure hysteresis effect in networks without astrocytic signaling and for weak cue stimuli (see Fig I in S1 Text). Note that the LPA-mediated signaling pathway modelled in this work is only one of a multitude of neuron-astrocyte and synapse-astrocyte communication pathways [29, 50]. To which degree other signaling pathways between synapses and astrocytes have similar or different effects on the network dynamics during WM remains an exciting topic for future research.

Our findings suggest that the slow timescale of LPA is crucial for its effect as an integrator (and thereby stabilizer) of recent WM activity. If LPA would unbind from the presynaptic receptors more quickly, the amount of presynaptic LPA would oscillate with the population spike and fail to ramp up over time, loosing the delayed, long-term effects on the calcium binding rate that form the basis of LPA-mediated modulation of transient WM termination in our model. We can see this effect in the distribution of durations that we obtain for very low LPA unbinding timescale $\tau_L$: the distribution can be matched with an analogous distribution in the STP network without LPA signaling, and neither shows broadening nor bimodality. Whilst the LPA unbinding timescale has a long-term stabilizing effect on synaptic WM, there are three reasons why it does not cause the WM dynamics to get stuck. First, even in the case of strong LPA signaling, the LPA-stabilized persistent WM representations can be terminated by a single, weak off-stimulus, just like in the case of regular persistent WM representations. Second, since the activity-dependent ramping of LPA is similarly slow as the decay of LPA-mediated effects, there is no limitation to follow faster WM protocols. For a fast WM protocol, the slow effects of LPA signaling would probably be only marginally observable since they start showing effect only several seconds after the cue offset. Third, even very long or repeated presentation of a WM stimulus does not affect the WM activity of subsequently presented, very different WM stimuli (i.e. with no overlap between encoding populations). This is because the amount of presynaptically bound LPA only ramps up within the stimulus-encoding

population, and, due to the limitation of LPA binding slots, only increases up to its bounded equilibrium value. As a result, it does not affect the activity in the remaining network after termination of WM activity. It is possible, however, that LPA ramping helps stabilize WM activity of very similar, subsequently presented stimuli or tasks (i.e. with a significant overlap of representations). The verification of this hypothesis and the detailed analysis of network behaviour in a multi-item WM setting with LPA signaling remains to be tested in future work.

Several extensions to our model can be made that lie outside the scope of this study. Importantly, we currently do not include astrocyte-to-astrocyte interactions in our model. Instead, we make a 'mean-field' assumption regarding the interactions between neurons and astrocytes, i.e. we assume that the astrocytic modulation of presynaptic STP dynamics happens independently in each synapse and in a statistically identical fashion throughout the network. Recent work by Gordleeva et al. [54] addresses the effects of astrocytic coupling during astrocyte-modulated WM formation. They model astrocyte-to-astrocyte interactions as diffusive coupling of astrocytic intracellular IP3 and calcium concentrations in a lattice of partly overlapping astrocytes. Similar to presynaptic LPA binding in our model, their astrocytic calcium and IP3 dynamics are dependent on synaptic activity. In contrast to our study, however, they do not include short-term synaptic plasticity in their model. Another exciting direction for future research is to apply our model in a multi-item WM setting similar to the settings explored for synaptic WM with STP [5, 67]. Due to the relatively fast decay of the STP traces, multiple WM items (each encoded by a separate neuronal subpopulation) are kept in memory by sequential firing of a single population spike from each population. We can speculate that, for the case of non-overlapping parallel WM items, the long timescale of astrocytic modulation could allow each memory to be active for an extended time before the next WM item has to be refreshed in memory. This intuition is supported by a preprint [68] that studied multi-item WM in a network with astrocytic modulation of STP. In that network, each astrocyte controls one synaptic ensemble that is involved in encoding a WM item [68]. In contrast to our study, they do not consider transient WM activity but only persistent and silent WM representations. For the case of partially overlapping, simultaneously active WM representations, the effect of the slow astrocytic modulation is less clear and is subject of future research. In particular, we can hypothesize that astrocytic signaling could also lead to fusion of overlapping WM representations, depending on the synaptic and astrocytic parameters. Another interesting topic to be explored in the future is the question how pathologies related to astrocytic signaling could impact WM (see section 'Potential impact of PRG1-associated pathologies on WM' in S1 Text).

Astrocytic LPA-signaling allows for two possible mechanisms of top-down volitional WM control (illustrated in Fig 7). First, astrocytes can affect the duration of WM activity after cue presentation (see Figs 2, 4 and 5). Each astrocyte covers around $10^5$ synapses in mice and $2 \times 10^6$ in humans [38], and could therefore affect LPA signaling in this subpopulation of synapses via the modulation of ATX. The second pathway is heterosynaptic: Incoming synaptic current of neighboring synapses from the thalamocortical loop could lead to a postsynaptic surge in calcium, thereby temporarily inhibiting PRG1 activity [53]. In consequence, LPA concentrations in the synaptic cleft would rise and allow more LPA to bind to its presynaptic receptors, reflected by an increase of the parameter $M$. This could lead to a change of the properties at the synapse, making WM memories more robust to noise. Both mechanisms constitute candidate pathways for the volitional control of WM [12, 69, 70], e.g. from the beta-band activity of deep layers to superficial layers [69]. A previously proposed type of volitional WM control through astrocytic modulation of STP suggests that astrocytes could switch the WM regime from silent into persistent mode [68]. The type of volitional control we propose here is different: it affects WM more subtly by modulating the duration and stability of active representations. Both types address equally important aspects of volitional WM control, and we can

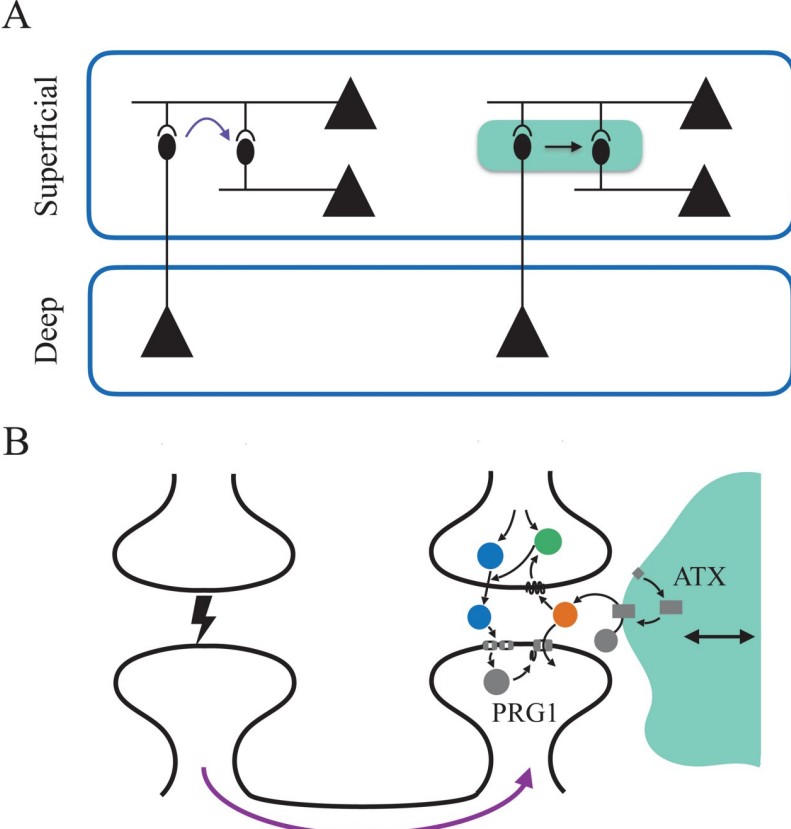

**Fig 7. Possible synaptic top-down control of working memory representations via the tripartite synapse. A** Deep cortical layers could drive oscillation in superficial layers, which represent specific working memories [69]. LPA signaling suggests two possible indirect pathways for such a control mechanism: Incoming activity from deep layers may contribute to PRG1 inhibition through heterosynaptic effects (left), or astrocytic signaling may increase LPA synthesis through astrocytic ATX, therefore modulating STP in recurrent connections (right). **B** Details of the heterosynaptic and astrocytic signaling pathway shown with magenta arrow and green shaded astrocytic cell, respectively.

speculate that neuronal networks in different brain regions would either (i) implement volitional WM control of one or the other type, depending on their neuronal and synaptic architecture; or (ii) switch between them depending on the level of surrounding global brain activity.

In practice, the modulation of WM introduced by LPA signaling could have several functions. First, since the variability of WM durations is dependent on LPA parameters in the weak astrocytic signaling case (Fig 5), our work predicts that the available 'precision' of WM durations could differ across tasks or brain structures—and hence allow for more or less spurious behavioural regimes. In contrast, for the case of strong astrocytic signaling, we observe a switch-like behaviour between very short and very long WM durations (filtering) without an increase in variability. While we show which synaptic/astrocytic and network mechanisms would be responsible for the spuriousness of WM, our work by itself does not allow to draw conclusions on which behavioural regime would be used by animals or humans during specific tasks or in specific brain regions. Second, the 'window of vulnerability' that is introduced by strong LPA signaling effects on presynaptic STP could implement a filtering of WM representations. During this initial 'window of vulnerability', the WM representation is very susceptible

to disruption by noise, such that WM representations with slightly higher internal noise would have a high chance to terminate during this period. Such slightly elevated levels of noise could e.g. arise from the presence of distractors or more salient/attended and therefore stronger competing WM representations in the same network. We are currently not aware of experimental studies explicitly testing the impact of distractors that are presented at different time points during the delay period, such that the experimental verification or falsification of this prediction remains the topic of future research. The WM representations that survive the initial 'window of vulnerability' are then stabilized for a longer time to allow for further cognitive processing. The filtering implemented by this mechanisms could be relevant for prioritizing and managing the limited mental capacity to hold WM items. In particular, it opens up the possibility of competitive processes between ambiguous stimuli, where multiple factors such as the order of presentation or the salience of a WM item could be important for which WM representation will be the 'winning' one that stays active. In that sense, the 'window of vulnerability' could also be interpreted as a 'window of opportunity' to resolve ambiguous activity. At the same time, the filtering due to strong astrocytic signaling could help to save energy by limiting the duration of WM representations that do not require further processing and therefore do not need to be active for a long time—especially since the slow unbinding of LPA allows them to be reactivated from their synaptic traces for a much longer time than in the absence of astrocytic modulation.

To conclude, our model explains how signaling in the tripartite synapse could influence the network response to a working memory cue. In particular, we show how slow astrocytic signaling can mechanistically implement a 'window of vulnerability' that limits the duration of metastable active working memory representations.

## Supporting information

**S1 Text. Supplementary text, figures and model descriptions.** We provide additional computations, numerical simulations and descriptions of the network models and their parametrizations.
(PDF)

## Acknowledgments

We would like to thank Johannes Vogt and Heiko Endle for valuable discussions and insights into the LPA-signaling mechanism, and Pierre Ekelmans for stimulating discussions and constructive feedback.

## Author Contributions

**Conceptualization:** Andreas Nold, Tatjana Tchumatchenko.

**Data curation:** Sophia Becker, Andreas Nold.

**Formal analysis:** Sophia Becker, Andreas Nold.

**Funding acquisition:** Tatjana Tchumatchenko.

**Methodology:** Sophia Becker, Andreas Nold.

**Project administration:** Tatjana Tchumatchenko.

**Software:** Sophia Becker, Andreas Nold.

**Supervision:** Andreas Nold, Tatjana Tchumatchenko.

**Validation:** Sophia Becker, Andreas Nold.

**Visualization:** Sophia Becker, Andreas Nold.

**Writing – original draft:** Sophia Becker, Andreas Nold, Tatjana Tchumatchenko.

**Writing – review & editing:** Sophia Becker, Andreas Nold, Tatjana Tchumatchenko.

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
