## [Decision Letter · Decision Letter 0]

17 Mar 2022

Dear Prof. Tchumatchenko,

Thank you very much for submitting your manuscript "Synaptic and astrocytic modulation of working memory durations" for consideration at PLOS Computational Biology.

As with all papers reviewed by the journal, your manuscript was reviewed by members of the editorial board and by several independent reviewers. In light of the reviews (below this email), we would like to invite the resubmission of a significantly-revised version that takes into account the reviewers' comments.

In addition to the comments related to the technical aspects, we expect the revised version to take into account the demands for a consequential reshaping of the structure of the manuscript so as to improve its organization and the way the results are presented.

We cannot make any decision about publication until we have seen the revised manuscript and your response to the reviewers' comments. Your revised manuscript is also likely to be sent to reviewers for further evaluation.

Sincerely,

Hugues Berry

Associate Editor

PLOS Computational Biology

Samuel Gershman

Deputy Editor

PLOS Computational Biology

Reviewer's Responses to Questions

**Comments to the Authors:**

Reviewer #1: The paper presents a consideration of a biologically motivated slow astrocytic regulation of presynaptic release probability in a working memory model. Using firing rate and spiking network simulations, the authors show how the duration and stability of working memory representations are altered by astrocytic regulation and by noise. It is a topic of interest to the researchers in the related areas.

For the readers, however, a number of points need clarifying and certain statements require further justification. My detailed comments are as follows:

(1) Abstract could be more specific. Please emphasize in the abstract Background, Methodology/Principal Findings, and Conclusions/Significance.

(2) My other more general comment is that the text needs some work. It’s quite difficult to read through long sentences. Especially in sections “Transient WM activity emerges from silent and persistent network dynamics due to noise” and “LPA signaling modulates the distribution of transient durations at the transition from the silent to the persistent state”.

(3) Why was such a number of neurons chosen? What determines the ratio of the number of neurons in selective and unselective populations? How will the behavior of the network change if inhibitory neurons are excluded? What topology of the astrocytic network and the structure of the organization of neuron-astrocytic interaction at the network level is assumed in the model?

(4) How the loading of an item into WM was designed? It is necessary to clarify whether the cue stimulation differed from stimulation parameters during training. How the system will behave in multi-item WM tasks? What dynamic mode of the system will correspond to free (not cued) recall?

(5) Please include in the introduction or discussion a brief review of the related model studies about neuron-astrocytes interaction in the WM formation (DOI: 10.3389/fncel.2021.631485; https://www.biorxiv.org/content/10.1101/2021.03.25.436819v1;
doi.org/10.1007/s00521-022-06936-9)

(6) It is worth noting that the paper format needs to be carefully edited, and some language expression methods of the paper need to be appropriately adjusted and polished.

Some minor points:

- Lines 60-61 Please provide the ref for “This is motivated by a recently described signaling pathway that involves the astrocytic and postsynaptic component, acts on each synapse individually, and introduces a slow timescale.”

- 143 it's not clear what the expression means “broadly stable transition”

- 181 the correct term is an Andronov-Hopf bifurcation.

- 190 please clarify the phrase “saddle node of the periodic solution”

- 237 abbreviation “WT” used before its introduction

- Fig.5A axis label “secs”

- Fig. 5B,C missing legend for yellow color. These figures are difficult for the reader to understand. Maybe they can be presented in a different way.

Reviewer #2: Becker et al.’s study is an elegant attempt to understand the biophysical mechanisms underpinning working memory (WM) duration. They present original arguments on how the combination of different noise sources could have opposite effects on WM encoding, either stabilizing or destabilizing it. Furthermore, they provide analytical and computational arguments that pinpoint astrocytic regulation of basal synaptic release as a critical factor in regulating the duration of WM neuronal correlates. Their modeling work extends the network model originally introduced by Mongillo and collaborators in Science 2009 to account for different noise sources and astrocytic regulation of presynaptic release by astrocytically-synthetized lysophosphatidic acid (LPA).

While the paper bears excellent potential for novelty and introduces exciting ideas, the current manuscript appears to be detrimentally disorganized. It is often hard to understand what model between spiking and rate model one is looking at. The exposition of the results could be substantially improved and currently suffer from figures only marginally explained. The questions raised in the introduction do not match the arguments presented in the discussion. Overall the manuscript should be self-contained, while it is not. Hence I am advising for major reshaping.

Please refer to the attached document for details.

Reviewer #3: The authors study the effect of lipid modulation on the time scales of short-term plasticity in a model of working memory. They use simulations in a spiking model and analysis using a firing rate model. They show how the modulation may influence WM duration, enhance WM robustness to noise (either preventing spurious induction or termination), and they discuss possible roles in top-down modulation of WM.

The article is well written and work appears to be scientifically correct. The authors has made a very good effort to make the models understandable by providing the information according to Nordlie.

Major points:

The problems in working memory that you address and your solutions to these problems need to be provided throughout your text (abstract, introduction). Consider your points about your main findings raised in the Discussion and formulate your research problem around this. More specifically see 1-3 below:

1) On line 142-142 you provide a good explicit statement. Please follow this and make similar statements for 2) and 3) below.

"In contrast, in the networks with LPA signaling, a slow and broadly stable transition from the silent to a transient regime is possible."

2) Section "Transient WM activity emerges from silent and persistent network dynamics due to noise".

You provide observations. Please remind the reader of what "problem" you want to address? Please provide explicit statements as outlined above.

3) Furthermore, in the section "LPA signaling modulates the distribution of transient durations at the transition from the silent to the persistent state"

Make more explicit statements of what your model with LPA-modulation results in, what "problem" do you address and what effect/solution do you observe?

Induction and maintenance or WM studied has been extensivley, including tolerance to noise or distracters. However, termination of activity has been addressed to a lesser extent. In this regard, you mention volitional control of WM which I think is an important point. There are two sides of the coin, providing stability to perturbations but at the same time allowing a possibility to end working memory. With regard to ending the activity, even if animal experiments of WM has shown results for delay periods even up 60 seconds, typically time scales of just a few seconds are used both for delay periods and inter-trial intervals, see e.g. this study on humans:

Teki Sundeep, Griffiths Timothy D. Working memory for time intervals in auditory rhythmic sequences. Frontiers in Psychology 5, 2014, DOI=10.3389/fpsyg.2014.01329

This issue of short time scales also includes continuous versions of the task (cDNMS) used in rodent experiments where working memory has to be continiously updated, e.g.

Young BJ, Otto T, Fox GD, Eichenbaum H. Memory representation within the parahippocampal region. J Neurosci. 1997;17(13):5183-5195. doi:10.1523/JNEUROSCI.17-13-05183.1997.

Here is finally my question: Your model adds a slow time scale (seconds-minute). Is there any risk of getting "stuck" in a too stable or hard-to-terminate state? Does this slow time-scale prevent you from following faster-changing working memory protocols?

Minor points:

Figure 1A.

Complement the signalling network figure 1A to indicate enhancing (+) or suppressive (-) effects on targets (LPA2r, ATX, PRG1, "active zone") via messengers (LPA, Ca2+, Glu) and indicate effect on targets. Specifically separating effects on messenger concentration or time constant of some process. Also illustrate your minimal model implementing this signalling (eq 4,5).

line 103-104

"the LPA-mediated effects in the presynapse occur on longer timescales of several seconds to a minute [45]."

Please explain how you arrived at this time scale for LHA. As far as I understand the article [45] discusses Ca2+ transients on that time scale.

Regarding reference to

Sreenivasan, K.K., D’Esposito, M. The what, where and how of delay activity. Nat Rev Neurosci 20, 466–481 (2019). https://doi.org/10.1038/s41583-019-0176-7

Since you cite this article, it would be suitable to also cite the modeling work discussed:

Pittà, M. D., De Pittà, M., Ben-Jacob, E. & Berry, H. Astrocytic theory of working memory. BMC Neurosci. 15 (Suppl. 1), P206 (2014).

Please discuss any similarities or differences between this model and your model.

Fig S1.

Your figure shows ISIs of 50, 100 and 200 ms. You provide a reference to ref 44. This artile uses the ISI of 500 ms. Please explain the data behind your figure.

Ref 41, 44

Both these references report work on hererozygous knock-out animals. This means that only one gene was deleted and the other gene was left functional. Naively speaking, this would render the protein content at 50%, so it would not mean the protein is down to 0%. Does that affect your model an any way?

**Have the authors made all data and (if applicable) computational code underlying the findings in their manuscript fully available?**

Reviewer #1: Yes

Reviewer #2: **No: **The authors guarantee that the data and code for their simulations will be made available once the manuscript will be approved for publication.

Reviewer #3: Yes

PLOS authors have the option to publish the peer review history of their article (what does this mean?). If published, this will include your full peer review and any attached files.

Reviewer #1: No

Reviewer #2: No

Reviewer #3: No
---

## [Decision Letter · Decision Letter 1]

22 Jul 2022

Dear Prof. Tchumatchenko,

Thank you very much for submitting your manuscript "How synaptic and astrocytic mechanisms can modulate working memory durations" for consideration at PLOS Computational Biology. As with all papers reviewed by the journal, your manuscript was reviewed by members of the editorial board and by several independent reviewers. The reviewers appreciated the attention to an important topic. Based on the reviews, we are likely to accept this manuscript for publication, providing that you modify the manuscript according to the review recommendations.

Sincerely,

Hugues Berry

Associate Editor

PLOS Computational Biology

Samuel Gershman

Deputy Editor

PLOS Computational Biology

[LINK]

Reviewer's Responses to Questions

**Comments to the Authors:**

Reviewer #1: Dear authors,

Thank you for addressing all of my comments.

Reviewer #2: I appreciate the substantial effort put by the authors in editing their manuscript. I found it remarkably improved in clarity and quality in its new version. There are still some issues that I believe require minor-to-moderate editing. I am detailing my suggestions for editing below, referring to the manuscript with tracked changes provided in the supplementary. I may anticipate these authors’ frustration with yet another round of reviews. However, I want to reassure them that it is the last one.

Please refer to the attached document for my detailed suggestions for edits.

Reviewer #3: I think the authors have provided strong responses to my questions and I find the manuscript has been greatly improved. I only have some remaining points for authors to think about in a larger picture and decide how they want to interpret their results and put these into an experimental context.

Regarding my question in the previous round on whether the long time scale integration could result in the network becoming "stuck" in the persistent state, I am satisfied with the answers from the authors regarding the persistent state but I have a follow-up question. If we take the example I brought up with the cDNMS task where the animal is presented with a long sequence of items, could your long time scales lead to a slow accumulation of your quantity l? See for instance your new text in Discussion, line 417 "We can think of LPA as a slow integrator of past synaptic activity." Suppose the cDNMS task presented to the animal changes, lets say to a task where dynamics stemming from high l would not be beneficial, would the slow kinetics not prevent a rapid and flexible adaptation to the new requirements and instead leaving the animal behaviourally "stuck" in the high-l-state? Any biochemical pathways leading to the (rapid) breakdown of LPA would open up for a decrease of tau_L and resolve this issue I presume.

You study how noise can affect and potentially control the activity in the metastable regime. I subscribe to the notion of the brain as an information processing system, and in this regard it is unclear why deepely located structures supporting WM would be transmitting or producing pure noise. Whereas the properties of the noise and your implementation of it in the model is clear, it is thus unclear to me how we should understand this "noise" from a biological perspective. You provide examples of how calcium and synaptic transmission appears to be guided by fluctuations and this seems fine. However, when it comes to neuronal spike fluctuations, are these really to be seen as "noise"? Neuronal firing contains a lot of correlations by virtue of neurons in a connected network. Moreover, what differentiates this activity you call noise from activity in your background activity or your cue activity?

Line 317 "longer transient WM representations therefore come with a higher variability of the durations"

Your model thus predicts that when you go from neglibile to low to medium LPA signalling, you would observe increasing variability of the length of the transients. Behaviourally this seems to indicate a rather spurious behaviour emerging, which seems an unfavourable functional mechanism. Please put your functional prediction into a behavioural context.

Regarding the "window of vulnerability", line 359-360. Firstly, are you predicting that the influence of distractors would be more influential shortly after presentation of the cue and instead distractors would become less potent after some amount of time? Are you aware of any experiments suporting this prediction?

Secondly, maybe we can interpret this feature differently? I find your new text in the Discussion intriguing, lines 481-483 discussing how more salient or more attended stimuli might be stronger when competing with other WM representations. I find this intriguing as it opens up for competitive processeses following e.g. ambigious stimuli where perceptual rivalry leads to one "winning" representation. So, instead of a "window of vulnerability" this could be viewed as a "window of opportunity" to resolve ambigious activity. Mechanisms that could modulate a competitive process including multiple factors such as which pattern appeared first or which is more salient, would be important.

Regarding your new comment on multiple items, I am not convinced it holds. You do use the careful wording "by a separate neuronal population", line 452-453, and "separate" is the key word here. If representations of different items overlap, activation of the items may result in the two representations not staying separate but becoming merged, see for instance Warden and Miller, Cerebral Cortex, 2007. As much as I understand you wish to provide a positive response to the question of the reviewer, this might be a slippery slope.

Minor:

Figure 2A.

On the y-axis, You use the unit Hz for "firing rate". I more often see Hz used for firing frequency (time or spatially averaged activity) and s^(-1) for rate (instantaneous firing frequency).

Figure 7.

"Incoming current ..." what kind of "current" is this? Maybe "activities"?

**Have the authors made all data and (if applicable) computational code underlying the findings in their manuscript fully available?**

Reviewer #1: None

Reviewer #2: Yes

Reviewer #3: Yes

PLOS authors have the option to publish the peer review history of their article (what does this mean?). If published, this will include your full peer review and any attached files.

Reviewer #1: No

Reviewer #2: No

Reviewer #3: **Yes: **Erik Fransén

Figure Files:

Data Requirements:

Reproducibility:

References:

---

## [Editor Report · Decision Letter 2]

5 Sep 2022

Dear Prof. Tchumatchenko,

We are pleased to inform you that your manuscript 'Modulation of working memory duration by synaptic and astrocytic mechanisms' has been provisionally accepted for publication in PLOS Computational Biology.

By then, please take care to change the title of the second section of the Results to "Noise can turn activity-silent and persistent firing regimes into transient WM activity", as proposed in your answer to the reviewer 2 (I could not find this change implemented in the revised manuscript)?

Best regards,

Hugues Berry

Academic Editor

PLOS Computational Biology

Samuel Gershman

Section Editor

PLOS Computational Biology

---

## [Editor Report · Acceptance letter]

27 Sep 2022

PCOMPBIOL-D-21-02324R2 

Modulation of working memory duration by synaptic and astrocytic mechanisms

Dear Dr Tchumatchenko,

I am pleased to inform you that your manuscript has been formally accepted for publication in PLOS Computational Biology. Your manuscript is now with our production department and you will be notified of the publication date in due course.

With kind regards,

Anita Estes
